# Spherically Stratified Point Projection: Feature Image Generation for Object Classification Using 3D LiDAR Data

**DOI:** 10.3390/s21237860

**Published:** 2021-11-25

**Authors:** Chulhee Bae, Yu-Cheol Lee, Wonpil Yu, Sejin Lee

**Affiliations:** 1Department of Mechanical Engineering, Kongju National University, Cheonan 31080, Korea; bch3494@kongju.ac.kr; 2Artificial Intelligence Laboratory, ETRI, Daejeon 34129, Korea; yclee@etri.re.kr (Y.-C.L.); ywp@etri.re.kr (W.Y.); 3Department of Computer Software, University of Science and Technology, Daejeon 34113, Korea

**Keywords:** spherically stratified point project, feature image, semantic labeling, point cloud

## Abstract

Three-dimensional point clouds have been utilized and studied for the classification of objects at the environmental level. While most existing studies, such as those in the field of computer vision, have detected object type from the perspective of sensors, this study developed a specialized strategy for object classification using LiDAR data points on the surface of the object. We propose a method for generating a spherically stratified point projection (*sP*${}^{2}$) feature image that can be applied to existing image-classification networks by performing pointwise classification based on a 3D point cloud using only LiDAR sensors data. The *sP*${}^{2}$’s main engine performs image generation through spherical stratification, evidence collection, and channel integration. Spherical stratification categorizes neighboring points into three layers according to distance ranges. Evidence collection calculates the occupancy probability based on Bayes’ rule to project 3D points onto a two-dimensional surface corresponding to each stratified layer. Channel integration generates *sP*${}^{2}$ RGB images with three evidence values representing short, medium, and long distances. Finally, the *sP*${}^{2}$ images are used as a trainable source for classifying the points into predefined semantic labels. Experimental results indicated the effectiveness of the proposed *sP*${}^{2}$ in classifying feature images generated using the LeNet architecture.

## 1. Introduction

Object recognition in autonomous navigation relies on various deep-learning methods to perform tasks such as detection and classification, which are being actively investigated [1,2]. The safety of the path traveled by an autonomous vehicle and its ability to avoid obstacles depend on the accurate classification of objects surrounding the vehicle [3]. Apart from autonomous driving, object classification is necessary for various applications and represents the basis for object recognition [4,5].

Owing to the accessibility of data, most object-classification methods use images collected through vision sensors. A vision sensor, however, is greatly influenced by environmental factors such as lighting and weather conditions. Consequently, unstable performance may be observed in autonomous vehicles that rely only on image data, leading to insufficient reliability for deployment in real scenarios [6,7]. Therefore, object recognition based on LiDAR sensors, which are less sensitive to environmental factors than vision sensors, must be studied together [8].

Various deep-learning methods have been applied for object classification using only data from LiDAR sensors. The performance of such methods depends on distance measurements from LiDAR sensors and the input data used for learning [9,10]. In fact, the training performance of a deep-learning algorithm varies according to the input data, and the quality and quantity of the input data can substantially affect the learning results [11]. Therefore, range measurements from LiDAR sensors should be complemented by an effective method to generate input data and achieve high-performance object classification based on deep learning. In object classification using vision sensors, several representative deep-learning algorithms have been devised to operate only with input images [12,13]. Thus, an available training image dataset can be directly used in existing high-performance algorithms to perform object classification. In contrast, in the case of object classification based on LiDAR sensors, it is difficult to generate a large amount of training data by performing distance measurement in the target environment, and the amount of open training data is also less than that of the visual image. Furthermore, raw LiDAR data cannot be used directly in deep-learning algorithms that use visual images. These problems can be solved by designing network architectures optimized for the sparse three-dimensional (3D) points provided by LiDAR sensors.

The direct use of LiDAR data for deep-learning methods has been proposed for object classification. For instance, object classification has been achieved using a 3D convolutional neural networks (CNNs) that use a 3D point cloud as the input using a volume representation through an occupancy probability update [14]. However, an inaccurate volume expression may be obtained owing to sporadicity in 3D point clouds, and the computational burden is high. Another method clusters the 3D points that remain after extracting ground data [15]. Although this method provides high segmentation speed and accuracy, segmentation is only achieved for distant objects, without providing meaningful classification. In [16], a learning network directly used the x-, y-, and z-axis coordinates of a randomly distributed 3D point cloud as the input data. The network achieved a high performance for simple object classification and real room segmentation, and the improved model [17] extracts feature vectors containing local information to improve accuracy. In addition, labeled voxels from a 3D point cloud map have been obtained for the training of a 3D CNN [18,19]. This method aims to classify several types of urban environment objects and exhibits accurate segmentation. In addition, studies have been conducted to apply the Hough transform as an image transformation method to the 3D point cloud. The trunk part of the trees was detected by classifying the point cloud according to the height and converting it into a binary image [20]. Another study was performed to detect a plane in a LiDAR point cloud using the Hough transform, which is faster than random sample consensus (RANSAC) [21]. The point clouds corresponding to the different objects were simply imaged by the Hough transform and classified by the CNN learning method [10]. In addition, many researchers have studied the segmentation of point clouds using the labeled point cloud dataset SemanticKITTI [22]. SPVConv [23] is a lightweight 3D module specialized for small object recognition to improve the recognition performance of small objects. In [24], a 3D-point-cloud-representation method called cylinder partition was designed for the LiDAR point segmentation of autonomous driving scenes and used with a 3D-convolution-based network. The range–point–voxel convergence network (RPVNet) [25] uses the UNet-based structure to compensate for the shortcomings of the voxel method in which information loss occurs due to lowering the resolution. In addition, SalsaNext [26,27] consists of an encoder-to-decoder architecture with a ResNet blockset in the encoder unit and upsampling in the decoder part in order to perform the semantic segmentation in an uncertainty-aware environment. Various other classification and segmentation methods for 3D point clouds have been proposed [28,29,30,31]. However, directly using LiDAR point cloud data in deep-learning methods remains challenging.

The above-mentioned reports proposed a unique learning network mainly for the classification of 3D point clouds. Even if a dedicated network is designed, the learning performance cannot be guaranteed according to the change of the input data. It is difficult to classify the individual points because learning is mainly performed in units of scenes. Overall, given the infeasibility of directly adopting state-of-the-art image-based deep-learning architectures, the use of LiDAR data is limited with regard to algorithm development. Even if a dedicated network is designed, the learning performance cannot be guaranteed for variations in the input data. Overall, deep learning using LiDAR data is limited regarding algorithm development given the infeasibility to directly adopt demonstrated deep-learning architectures that use images.

In this paper, we propose a spherically stratified point projection (*sP*${}^{2}$) method to generate feature images for processing 3D point clouds using existing deep-learning architectures. The proposed *sP*${}^{2}$ method can generate feature images for all 3D points by analyzing the distribution of surrounding points in a 3D point cloud. Thereafter, the generated feature image can be classified using an existing deep-learning method for two-dimensional (2D) images, thus achieving the pointwise classification of point clouds.

The *sP*${}^{2}$ feature images can provide unique geometric descriptions in terms of individual points from 3D data. Object classification using a general 3D point cloud differs from labeling through the clustering of points contained in an object. In addition, unlike existing independent networks, the *sP*${}^{2}$ feature images can be directly applied to architectures such as LeNet [32], GoogleNet [33], and AlexNet [34], which are widely used for image classification. To validate the proposed *sP*${}^{2}$, we conducted an experiment to classify urban structures using data collected from the Kongju National University (KNU) campus using a Velodyne 16-channel LiDAR (the KNU dataset) and the KITTI [35] dataset.

Our main contributions are summarized as follows:We propose a feature-image descriptor, which includes geometric information, based on only 3D points collected through a LiDAR sensor;The generated feature images include distribution information such as the location, distance, and density of surrounding points near a target point;The proposed feature-image-generation method is applicable to all 3D point clouds and enables pointwise classification through the popular image classifiers such as the CNN model;The proposed *sP*${}^{2}$ method was validated through learning based on the feature-image-generation method, and image-classification networks are evaluated on the KNU and KITTI datasets.

The remainder of this paper is organized as follows: Section 2 describes the proposed *sP*${}^{2}$ method to capture feature images that can be used as the learning input data obtained from a 3D point cloud. In Section 3, we experimentally evaluate the effectiveness of the proposed *sP*${}^{2}$ method. Finally, we draw conclusions and discuss further works in Section 4.

## 2. Spherically Stratified Point Projection

Vision sensor data, which are the most common type of data in image classification, provide information on the color of an object. This is advantageous for image learning because color data convey more information on each pixel within a fixed neighborhood of pixels. However, data collected using LiDAR sensors may represent object information pertaining to its surface, rather than its color. For example, the walls of a building can be represented as flat surfaces, while tree trunks can be represented as cylinders. Based on these advantages of LiDAR sensors, we propose a feature-image-generation method to define the attributes of the point unit using the point cloud distribution and the surface information of the object. The $sP^{2}$ uses the point cloud data measured by LiDAR as the input, and it generates the feature images that can be used directly in the object classifier. The CNN model as the image classification can output the resulting classified points corresponding to the objects using the $sP^{2}$ feature images.

### 2.1. Image Descriptor

The *sP*${}^{2}$ image descriptor uses the distribution of surrounding points to define the features of each and every point in a 3D point cloud. The surrounding points are centered on the origin and are selected by referring to a three-layer imaginary sphere consisting of a triangular grid. Since the spherical space is divided into triangular grids, the space can be represented by many triangulation grids through splitting a regular sphere into a geodesic sphere. Basically, the sphere space cannot be divided only with other different shapes such as hexagonal grids or pentagons. For instance, the patches of a soccer ball should be composed of a combination of hexagons and pentagons. To define the surrounding points, the straight-line distance $P_{st}$ is transformed by the distance between the origin $P_{i}$ from which the image will be created and all input points $P_{1:n}$ into a straight-line distance as in: (1)$$P_{d(x,y,z)}=P_{1:n}-P_{i},\;P_{st}=\left|\right|P_{d(x,y,z)}\left|\right|;\;P_{st}<r,\;P\in P_{N},$$
where the straight-line distance $P_{st}$ is less than the radius *r* of the sphere and the point *P* should be included in the neighboring point group $P_{N}$. As shown in Figure 1, the selected neighboring points are projected onto the nearest triangular grid of the virtual sphere, and the distribution characteristics of neighboring points with respect to the origin are updated by calculating the occupancy probability of the projected triangular grid. Each grid has an occupancy probability that is updated by the parameters such as the distance between the points and the number of points in the grid. In the triangulation grid, the occupancy probability can arithmetically extract the surface characteristic of the objects from the 3D point cloud. After that, one feature image is generated by matching the updated occupancy probability to each image pixel.

### 2.2. Occupancy Grid Update

Let $P_{t}$ represent a point cloud at time step *t* and $\tau_{i}$ be the target point to be classified. $P_{t}$ moves with $\tau_{i}$ being the origin, as shown in:(2)$$X\left({\tilde{P}}_{t,i}\right)=X\left(P_{t}\right)-X\left(\tau_{i}\right),\;i\in n(P_{t}),$$
where ${\tilde{P}}_{t,i}$ represents the group of points moved when $\tau_{i}$ is the origin and *X* represents the Cartesian coordinates (x, y, z) of the corresponding point. By applying the Bayesian model [36], each grid of the geodesic sphere centered near an arbitrary origin can update its occupancy accumulation as follows:(3)$$p\left(M|{\tilde{P}}_{t,i}\right)=\prod_{n=1}^{N}p\left(m_{n}|{\tilde{P}}_{t,i}^{\left(n\right)}\right),$$
where *M* represents the grids segmented from the *N* grids constituting the geodesic sphere and ${\tilde{P}}_{t,i}^{\left(n\right)}$ denotes a partial point cloud included in the bearing angle boundary condition satisfying the *n*th patch $m_{n}$. The occupancy probability of $m_{n}$ can be expressed using Bayes’ rule as follows:(4)$$p{\left(m_{n}|{\tilde{P}}_{1:j}^{\left(n\right)}\right)}_{j=1:J}=\frac{p({\tilde{\tau }}_{j}^{\left(n\right)}|m_{n})p\left(m_{n}|{\tilde{P}}_{1:j-1}^{\left(n\right)}\right)}{p({\tilde{\tau }}_{j}^{\left(n\right)}|{\tilde{P}}_{1:j-1}^{\left(n\right)})},$$
where *j* denotes the index of every point belonging to ${\tilde{P}}_{t,i}^{\left(n\right)}$ from the *J* points and ${\tilde{P}}_{t,i}^{\left(n\right)}$ should be reduced to ${\tilde{P}}^{\left(n\right)}$. Let ${\tilde{P}}^{\left(n\right)}$ from one to *j* be denoted as ${\tilde{P}}_{1:j}^{\left(n\right)}$. When deriving the probability that the opposite case of (Equation 4) will occur, some probability terms that are difficult to calculate by dividing (Equation 4) are deleted, and finally, this probability is expressed as follows:(5)$$\frac{p(m_{n}|{\tilde{P}}_{1:j}^{\left(n\right)})}{p({\bar{m}}_{n}|{\tilde{P}}_{1:j}^{\left(n\right)})}=\frac{p\left(m_{n}|{\tilde{\tau }}_{j}^{\left(n\right)}\right)p\left(m_{n}|{\tilde{P}}_{1:j-1}^{\left(n\right)}\right)p\left({\bar{m}}_{n}\right)}{p\left({\bar{m}}_{n}|{\tilde{\tau }}_{j}^{\left(n\right)}\right)p\left({\bar{m}}_{n}|{\tilde{P}}_{1:j-1}^{\left(n\right)}\right)p\left(m_{n}\right)}.$$

The log odds ratio for (Equation 4) is defined as follows:(6)$$\begin{array}{cc} l_{j}\left(m_{n}\right)= & log\frac{p(m_{n}|{\tilde{\tau }}_{j}^{\left(n\right)})}{1-p(m_{n}|{\tilde{\tau }}_{j}^{\left(n\right)})}+log\frac{p(m_{n}|{\tilde{P}}_{1:j-1}^{\left(n\right)})}{1-p(m_{n}|{\tilde{P}}_{1:j-1}^{\left(n\right)})}-log\frac{p(m_{n})}{1-p(m_{n})}, \end{array}$$
where the first term on the right-hand side of the equation represents the point projection model, which indicates the grid occupancy probability update according to the distance between grid $m_{n}$ and point ${\tilde{\tau }}_{j}^{\left(n\right)}$. The second term represents the occupancy probability update before calculating the current occupancy probability update with point ${\tilde{\tau }}_{j}^{\left(n\right)}$. The third term is determined by the prior probability of the lattice as a log odds ratio. We set $p(m_{n})$ to a large constant because $p(m_{n})$ is considered to be zero.

The function-point-projection model implements probability function $p(m_{n}|{\tilde{\tau }}_{j}^{\left(n\right)})$ in the log odds form. This model is applied to all points within the bearing region of each grid. Each point contributes towards updating the occupancy probability of the corresponding grid according to distance as follows:(7)$$p\left(m_{n}|{\tilde{\tau }}_{j}^{\left(n\right)}\right)=1.0-\frac{0.5}{1.0+e^{-\alpha (r_{j}-\mu )}},$$
where $r_{j}$ denotes the distance of ${\tilde{\tau }}_{j}^{\left(n\right)}$ projected onto the corresponding grid surface, α denotes the slope of the sigmoid function, and μ is a parameter to control the allowed distance between points participating in the occupancy probability update. Figure 2 is an illustration of an example of updating the occupancy probability in (Equation 7), where $p(m_{n}|{\tilde{\tau }}_{\left(1,2,3\right)})$ is (0.986,0.75,0.508) with α=4, μ=1.5 m, and $r_{\left(1,2,3\right)}=\left(0.6,1.5,2.5\right)$ m for ${\tilde{\tau }}_{j}$. The point projection model function in the form of the log odds ratio in (Equation 6) becomes (1.848, 0.477, 0.014). When the lattice surface of the updated geodesic sphere is two-dimensionally flattened, a unique image containing the shape correction characteristics of the center is obtained.

### 2.3. Image Generation

The overall flowchart is shown in Figure 3 according to the sequences from LiDAR points to classified points as the input and output, respectively. This process includes the method to extract $sP^{2}$ feature images that can be directly applied to the CNN classification. In Algorithm 1, we set up several parameters, which include the radius values and the number of occupied triangulation grids. First, the radius values on the spherical domain were determined by considering the LiDAR specification and the classified objects. The experimental data were collected by using 16-channel LiDAR and setting humans as the smallest object to classify. According to these experimental conditions, the radius values of $r_{1}$, $r_{2}$, and $r_{3}$ were determined as 0.25, 0.5, and 0.75, respectively. In addition, the number of triangulation occupied grids was derived by the spherical resolution in the geodesic domain. We executed dividing the spherical region into one-degree resolution considering the distribution of LiDAR points, and the space could consist of 180 triangulation occupied grids. Then, a feature input image was constructed of 14 × 14 in size by adding 16 empty grids to the 180 occupied grids. It first generates a three-layer virtual sphere consisting of a triangular lattice and centered at the target point to generate a feature image, as shown in Figure 3. All points collected by LiDAR can be candidates of target points. Normally, the target points are assigned according to the order of the input points. If only one virtual sphere is considered, the contained points are projected equidistantly on the surface of the sphere, and the surface information of the object may be lost; therefore, we expressed the unique surface information of the object in the form of an imaginary, three-layer sphere centered at the origin, as shown in Figure 4. Based on their linear distance from the origin, points around it were then divided into three layers. The spheres marked as $layer_{B}$, $layer_{G}$, and $layer_{R}$ contained points within a radius of $r_{1}$ units, $r_{1}$ to $r_{2}$ units, and $r_{2}$ to $r_{3}$ units, respectively, to update the distribution information for points close to the origin, midway from the origin, and far away from the origin, in that order. The layered point was used as an input to calculate the occupancy probability of the belonging layer. Namely, the values of $layer_{B}$, $layer_{G}$, and $layer_{R}$ are a different concept from the B, G, and R context channel data of a normal image because they contain the object outlines according to the region of the physical distances. An image of the point cloud projection and occupancy update for each layer is shown in Figure 4. The points divided according to the distance are projected on the triangular grid of the nearest sphere, and the occupancy probability of the triangle grid is finally updated to a value between zero and one according to the distance between the points and the center of the triangle grid. Consequently, a unique image including surface information is generated by encoding RGB values of color images to the unique images from the outermost to the innermost images. An image generated using *sP*${}^{2}$ is shown in Figure 5. Because a feature image is generated by developing an imaginary sphere based on a triangular grid, the pixel position corresponding to the triangular grid indicates the direction of the surrounding points from the origin. In addition, the saturation of a pixel represents the probability of occupancy of each triangular grid of an imaginary sphere. Finally, because the RGB channel is defined by three layers of virtual spheres, it represents the distance from the origin.
**Algorithm 1:***sP*${}^{2}$ image generation.    **Input:** $P_{1:n}$(x,y,z), $grid_{R}$, $grid_{G}$, $grid_{B}$  ▹$P_{1:n}$(x,y,z) represents the coordinates of n point clouds collected from LiDAR                      ▹$grid_{R}$, $grid_{G}$, and $grid_{B}$ are the grid information corresponding to layers R, G, and B    **Output:** *sP*${}^{2}$ image with dimensions of W×H corresponding to $P_{i}\left(x,y,z\right)$    **Parameters:** $r_{1}=0.25$, $r_{2}=0.5$, $r_{3}=0.75$         ▹$r_{1}$, $r_{2}$, and $r_{3}$ are the spherical radii demarcating the three layers1:**for **i=1→n** do**2:    Calculate the distance from the origin to all the points: $P_{d(x,y,z)}$=$P_{1:n}-P_{i}$3:    Convert the Cartesian coordinates to linear distances: $P_{st}=\left|\right|P_{d(x,y,z)}\left|\right|$4:    **for** j=1→n** do**   // Comparison of $P_{st}$ and *r*5:        **if** $P_{st}$ < $r_{1}$ **then**6:           add $P_{st}$ to $layer_{B}$7:        **else if** $r_{1}$≤$P_{st}$ < $r_{2}$ **then**8:           add $P_{st}$ to $layer_{G}$9:        **else if** $r_{2}$≤$P_{st}$ < $r_{3}$ **then**10:           add $P_{st}$ to $layer_{R}$11:        **end if**12:    **end for**13:    Find the closest grid by comparing the x,y,z values included in $layer_{B,G,R}$ with the center coordinate $grid_{B,G,R}$14:    **for** k=1→180 **do**15:        Update the occupancy probability for 180 grids of $grid_{B,G,R}$ using the number of points and $P_{st}$ included in the grid16:    **end for**17:    Add 16 blank grids after each grid to resize the image to dimensions of W×H18:    Generate a three-channel image for each $P_{i}$ by applying the values of $grid_{R}$, $grid_{G}$, and $grid_{B}$ to the RGB layers19:**end for**

## 3. Experimental Evaluation

We conducted an experiment that was divided into three stages. First, we trained a neural network using the generated *sP*${}^{2}$ images. Second, the classification performance was quantitatively evaluated using the *sP*${}^{2}$ images generated from 3D points based on the manually written ground truth. Third, we segmented point clouds based on the classification results using *sP*${}^{2}$ images generated from raw data without using ground truths. Two datasets were used to quantitatively validate the *sP*${}^{2}$ image generation regarding classification and segmentation.

### 3.1. Datasets and Training Setup

Experimental data were collected in two ways. The first was the KNU dataset. This is a collection of 3D point clouds scanned by a LiDAR scanner and the dead-reckoning position measured by an inertial measurement unit (IMU) on a mobile robot platform. It contains the labeled 3D point cloud data corresponding to various objects on the campus site to research self-driving technologies, such as mapping, localization, object classification, path planning, etc. Another dataset was extracted from the KITTI raw dataset. Considering the information collected for an urban environment, we aimed to classify the 3D points according to the following class: person, car, tree, building, and floor. The datasets have different point densities and reflect various environments. The KNU dataset was collected at various locations and angles within the campus to cover various situations, and its point density is low because the LiDAR sensor has 16 channels. The KITTI dataset has not been collected in various environments, but it has a high point density given the 64-channel LiDAR sensor used for its collection, thereby rendering it suitable to distinguish objects far from the origin. Some differences between the two datasets are illustrated in Figure 6. In addition, the labeling work was performed manually using the raw 3D points collected by LiDAR since the trained model used not only the public KITTI dataset, but also our own KNU dataset.

We performed labeling to generate *sP*${}^{2}$ images using the collected 3D point clouds. We only labeled items that exhibited the characteristics of the objects. For the KNU dataset, 45,000 images were obtained, with 9000 images per class, and for the KITTI raw dataset, 15,000 images per class were generated, yielding a total of 75,000 images for training. We trained the network using these datasets. Each *sP*${}^{2}$ image generated for training has 14×14×3 pixels. The feature images were employed for training LeNet, which has one of the simplest CNN structures and is suitable for confirming the pure effectiveness of *sP*${}^{2}$, and the learning network was constructed using NVIDIA-DIGITS. In addition, in comparison with LeNet, we can expect to achieve a higher level of accuracy by using complex and sophisticated architectures such as GoogleNet or AlexNet.

### 3.2. Classification Performance

The classification performance from existing methods is listed in Table 1, and that obtained from the proposed *sP*${}^{2}$ method for the two datasets is listed in Table 2. The spherical signature descriptor (SSD) [37] and the modified spherical signature descriptor (MSSD) [38] were tested on 2000 images per class using data similar to those from the KNU dataset. The proposed *sP*${}^{2}$ was tested on 500 images per class of the KNU dataset. Compared to the MSSD, the accuracy using the proposed method improved by 11% for a person, 12.15% for a car, 3.65% for a tree, and 12.05% for a building.

In addition, the proposed *sP*${}^{2}$ was tested on 1000 images per class of the KITTI dataset. Compared with the KNU dataset, the accuracy improved by 0.4% for a person, 4.2% for a car, 4.6% for a tree, 1.1% for a building, and 0.6% for a floor. The results indicated that the accuracy of *sP*${}^{2}$ is higher than that of the MSSD, which does not use the surface features of the object. Considering the two datasets, we conclude that the density of point clouds obtained from the LiDAR sensors influences the accuracy of the *sP*${}^{2}$ method, achieving a higher accuracy on the KITTI dataset with high point density.

### 3.3. Raw 3D Point Cloud Classification

The classification results of raw 3D point clouds are shown in Figure 7 and Figure 8. The classification result showed a pattern similar to the semantic segmentation result because the classification was performed based on each point. The classes for objects are represented in different colors, such as purple for people, green for cars, blue for trees, sky blue for buildings, and yellow for floors. Table 3 summarizes the results of a KITTI data 3D point group segmentation network proposed in a previous report, whereas Table 4 summarizes the results of KITTI data 3D point group classification using the *sP*${}^{2}$. In the previous study, various objects were classified using Semantic KITTI, but this study performed an experiment to classify the five types of objects in order to use the KNU dataset specialized for the urban environment. Here, pedestrians and cyclists were grouped into a class known as person. In the case of person, lower intersection over union (IoU) results were obtained compared to previous studies. Pedestrians and cyclists were classified into the same class, and many misclassifications occurred for the trunks of trees. For cars, the results were lower than those of previous studies, and the classification results of trees revealed the lowest performance. By analyzing the classification results, we found that points included in objects having a cylindrical shape such as signposts were classified as trees or people. In addition, cars and trees entailed a low classification performance owing to confusion regarding the similar distribution shape of point clouds of leaves and cars. The classification results of buildings and floors exhibited substantially high performance compared to the classification results of other objects. This is the difference in performance when using the *sP*${}^{2}$ method, which generates a feature image using the distribution characteristics of neighboring points for each point. In the case of buildings and floors, it can be verified that characteristics such as planarity and verticality were well classified, resulting in high performance. In addition, it was verified that our experimental method, which attempted to classify all 3D points into five classes, exhibited lower classification performance for cars and trees. For the scene-segmentation experiment, we used only points including a radius of 30 m from the LiDAR to reduce the computational burden. Nevertheless, the processing time to generate the *sP*${}^{2}$ image was below 1 Hz with an AMD Ryzen 5 hexa-core. This confirmed that the real-time performance cannot be guaranteed without the use of GPU parallel processing. It is necessary to calculate five-hundred forty occupancy probabilities of three layers and one-hundred eighty grids to generate the *sP*${}^{2}$ image. At this time, as the number of points increases, the amount of ancillary calculations increases as well according to Algorithm 1. This means that it is hard to accomplish the real-time performance with only the CPU.

## 4. Conclusions and Further Works

We proposed the *sP*${}^{2}$ to produce training images from point clouds rather than designing a dedicated network for point cloud classification, thereby establishing a novel paradigm for 3D point cloud classification. Our approach differs from that of previous studies in that it classifies all point clouds and uses an already designed image classification network. The *sP*${}^{2}$ feature images provide information about the surface of an object and the distribution of surrounding points. To verify the performance of the proposed *sP*${}^{2}$ method, we conducted experiments on the existing KITTI and KNU datasets collected from Kongju National University. In the training stage, the classification performance was high, as shown in Table 2, but the actual scene segmentation was low. The experimental method classified only five types of objects and did not consider the various objects in the real world. This resulted in low precision, especially decreasing the mIoU value. However, even with a simple LeNet architecture, we could achieve the high recall values of 98.4, 81.2, 53.1, 68.6, and 95.2 for the five target objects, person, car, tree, building, and floor, respectively. This result shows the robust classification performance with the simple network by using *sP*${}^{2}$ feature images.

As further works, we expect to achieve high segmentation performance by diversifying classification objects and using deeper classifier networks. In addition, we will adopt the advanced technical methods to generate *sP*${}^{2}$ images using parallel processing in order to satisfy the real-time performance.

## Figures and Tables

**Figure 1 sensors-21-07860-f001:**
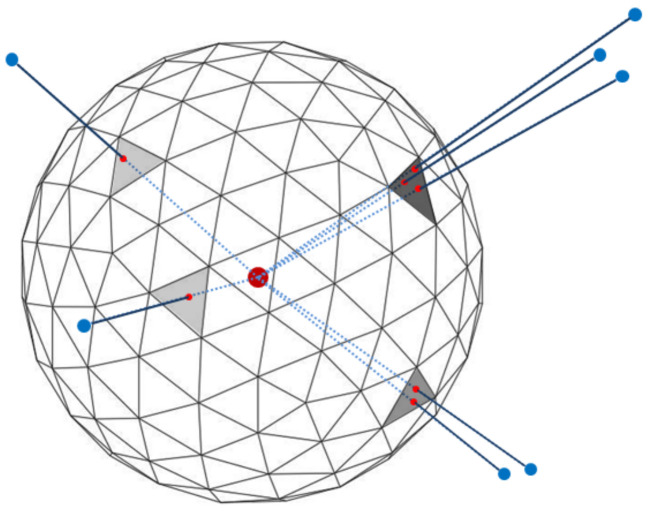
Geodesic tessellation of spherical surface and projection of neighboring points onto the corresponding grid. The large red dot in the middle of the sphere indicates the target 3D point to be classified into a semantic label. The blue dots indicate neighboring 3D points, which are projected onto the geodesic grid (small red dots).

**Figure 2 sensors-21-07860-f002:**
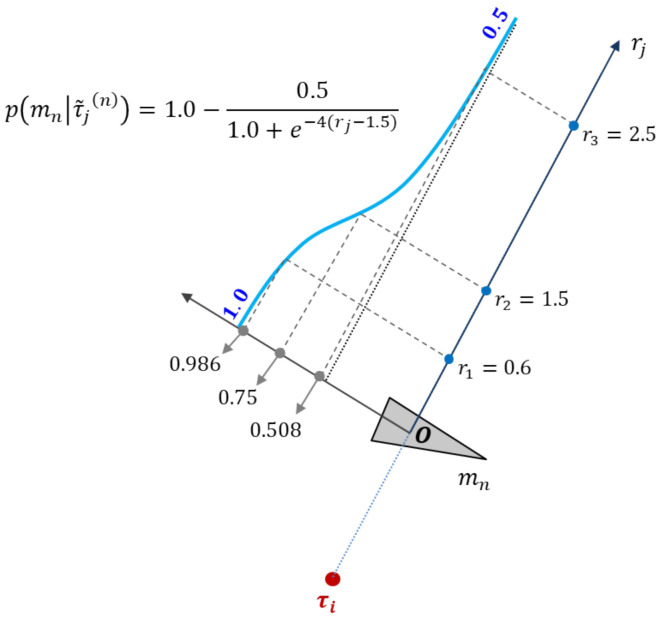
Example of updating the occupancy probability for α=4, μ=1.5 m, and $r_{\left(1,2,3\right)}=\left(0.6,1.5,2.5\right)$ m for ${\tilde{\tau }}_{j}$ in (Equation 7). The resulting $p(m_{n}|{\tilde{\tau }}_{\left(1,2,3\right)})$ is (1.848,0.477,0.014).

**Figure 3 sensors-21-07860-f003:**
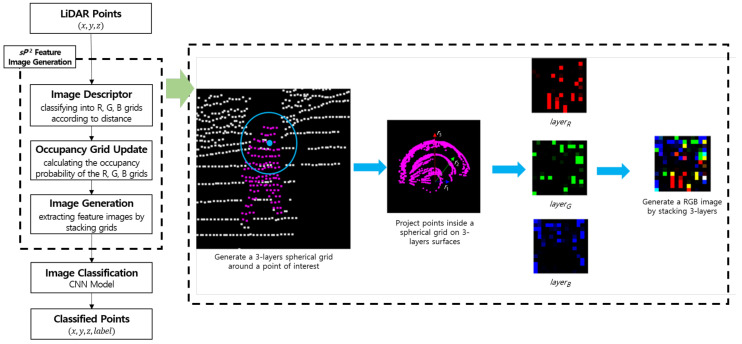
System flowchart the of LiDAR-point-cloud-based object classification method using the *sP*${}^{2}$ feature images.

**Figure 4 sensors-21-07860-f004:**
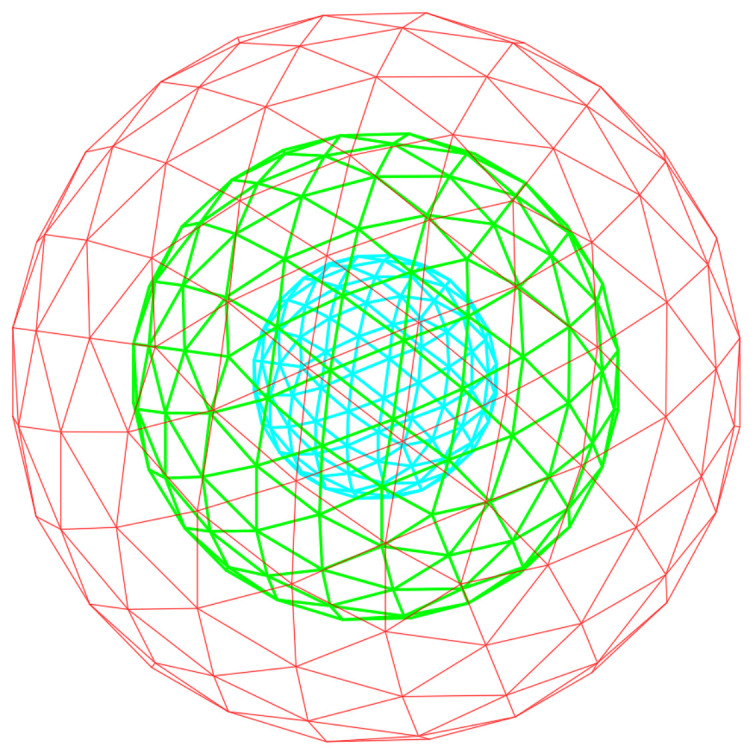
The outer sphere projects points far from the target point, and the points are assigned to red values in a color image. The middle sphere projects points are located midway from the target point, and the points are assigned to green values in a color image. The inner sphere projects points close to the target point, and the points are assignment to blue values in a color image.

**Figure 5 sensors-21-07860-f005:**
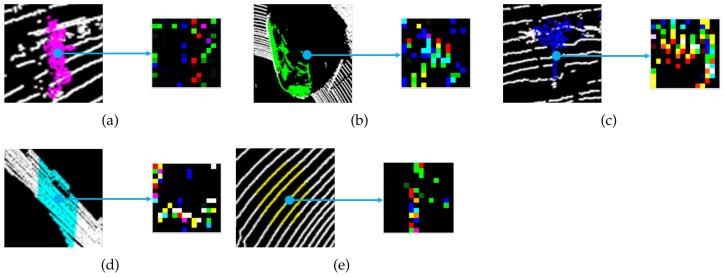
Example of a random *sP*${}^{2}$ image generated for various labels: (**a**) person, (**b**) car, (**c**) tree, (**d**) wall, and (**e**) floor.

**Figure 6 sensors-21-07860-f006:**
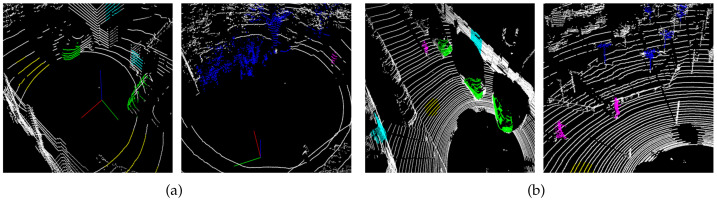
Raw data labeled by points of interest. Purple indicates persons, green cars, blue trees, sky blue buildings, and yellow floors. Samples from (**a**) the KNU dataset using 16-channel LiDAR acquisition and (**b**) the KITTI dataset using 64-channel LiDAR acquisition.

**Figure 7 sensors-21-07860-f007:**
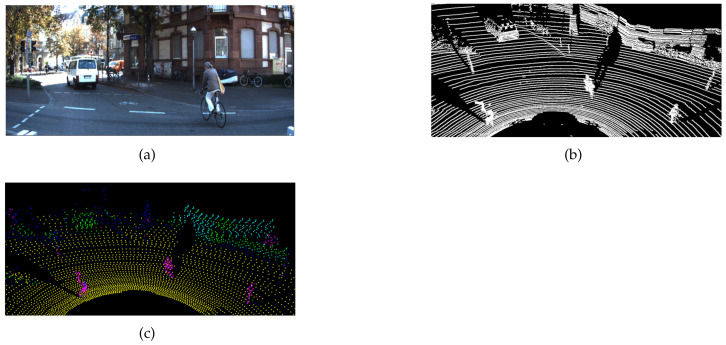
Segmentation evaluation on (**a**) the KITTI image and (**b**) the KITTI raw point cloud. (**c**) Point cloud segmented using the generated *sP*${}^{2}$ image.

**Figure 8 sensors-21-07860-f008:**
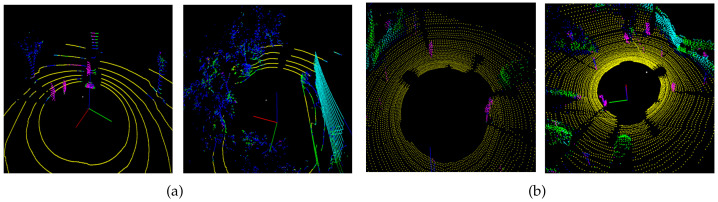
Segmentation of images from (**a**) the KNU dataset (16-channel LiDAR data) and (**b**) the KITTI dataset (64-channel LiDAR data).

**Table 1 sensors-21-07860-t001:** Classification accuracy of previous studies using the KNU dataset.

Method	Person	Car	Tree	Building	Floor
SSD [37]	-	90.1%	83.6%	91.7%	95.1%
MSSD [38]	88.2%	82.25%	90.65%	86.15%	-
*sP* ${}^{2}$	99.2%	94.4%	94.6%	98.2%	99.2%

**Table 2 sensors-21-07860-t002:** Classification performance according to learning using *sP*${}^{2}$ images for two datasets.

Dataset	Label	Person	Car	Tree	Building	Floor	Accuracy
	Person	496	2	2	0	0	99.2%
	Car	0	472	23	1	4	94.4%
KNU	Tree	5	10	473	10	2	94.6%
	Building	1	1	7	491	0	98.2%
	Floor	0	2	2	0	496	99.2%
	Person	996	0	4	0	0	99.6%
	Car	1	986	10	3	0	98.6%
KITTI	Tree	3	2	992	2	1	99.2%
	Building	0	2	5	993	0	99.3%
	Floor	0	1	1	0	998	99.8%

**Table 3 sensors-21-07860-t003:** Evaluation of the classification performance of previous studies.

Methods	mIoU	Car	Bicycle	Motorcycle	Truck	Other-Vehicle	Person	Bicyclist	Motorcyclist	Road	Parking	Sidewalk	Other-Ground	Building	Fence	Vegetation	Trunk	Terrain	Pole	Traffic Sign
PointNet [16]	14.6	46.3	1.3	0.3	0.1	0.8	0.2	0.2	0.0	61.6	15.8	35.7	1.4	41.4	12.9	31.0	4.6	17.6	2.4	3.7
SqueezeSegV3 [39]	55.9	92.5	38.7	36.5	29.6	33.0	45.6	46.2	20.1	91.7	63.4	74.8	26.4	89.0	59.4	82.0	58.7	65.4	49.6	58.9
SalsaNext [26]	59.5	91.9	48.3	38.6	38.9	31.9	60.2	59.0	19.4	91.7	63.7	75.8	29.1	90.2	64.2	81.8	63.6	66.5	54.3	62.1
Cylinder3D [24]	67.8	97.1	67.6	64.0	59.0	58.6	73.9	67.9	36.0	91.4	65.1	75.5	32.3	91.0	66.5	85.4	71.8	68.5	62.6	65.6
SPVNAS [23]	67.0	97.2	50.6	50.4	56.6	58.0	67.4	67.1	50.3	90.2	67.6	75.4	21.8	91.6	66.9	86.1	73.4	71.0	64.3	67.3
RPVNet [25]	70.3	97.6	68.4	68.7	44.2	61.1	75.9	74.4	73.4	93.4	70.3	80.7	33.3	93.5	72.1	86.5	75.1	71.7	64.8	61.4

**Table 4 sensors-21-07860-t004:** Classification performance evaluation of the *sP*${}^{2}$.

	mIoU	Person (Pedestrian+Cyclist)			Car (Car+truck)			Tree (Pole+Trunk+Vegetation)			Building (Wall)			Floor (Road)		
		Precision	Recall	IoU	Precision	Recall	IoU	Precision	Recall	IoU	Precision	Recall	IoU	Precision	Recall	IoU
*sP* ${}^{2}$	50.2	35.7	98.4	35.5	49.7	81.2	44.6	15.7	53.1	13.8	88.1	68.6	62.8	99.1	95.2	94.4

## Data Availability

Not applicable.

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
