# Peer review of "Spherically Stratified Point Projection: Feature Image Generation for Object Classification Using 3D LiDAR Data"

_sensors, 2021, doi:10.3390/s21237860_

Round 1
Reviewer 1 Report
The paper proposed a spherically stratified point projection method to extract local features o for semantic segmentation in LiDAR point clouds. The paper is well structured and the simulation is properly discussed. Some suggestions:
- The paper contribution is limited. Is there any contribution of your applied CNN model?
- Ablation experiment is suggested to be provided to explain the efficiency of the sP2 Meanwhile, the classification results should be compared with more typical neural networks for semantic segmentation.
- What’s the difference between Fig.1 and Fig.4? If redundent, please combine them into one figure.
- In Fig2, how can you determine which points in red is neighboring to the points in red?
Reviewer 2 Report
The idea presented in the submitted manuscript is promising with potential in real applications. However, there are several concerns that have to be clarified before publication.
- There are a lot of good object classification methods for LiDAR cloud points. The state-of-the-art results are not mentioned in the literature review. Moreover, the reader has the impression that there are not a lot of works in this area yet. So I recommend to update the Introduction section.
- Section 2.2 that describe occupancy grid update is not clear. Moreover, there is not know why it is used for. Is it anyhow utilize in encoding RGB values?
- RGB encoding suggests correlation with RGB color model. In my opinion, this correlation is not suitable in this case.
- Algorithm 1 is not clear. In particular: (a) how to choose the target point; (b) how to choose the values of the radios $r_1$, $r_2$ and $r_3$ - why values 0.25, 0.5 and 0.75 have been hardcoded in the algorithm?; (c) why updating the occupancy grid is performed for 180 grids and then 16 blank grids are added?
- There is no motivation behind the choose of the triangular lattice. Maybe a hexagonal one would be better?
- There is not clear how labelling of the collected 3D point clouds have been performed. Manually, based on camera image, semi-automatically? This is important as during this process the ground truth data are determined.
- There is lack of references to KITTI dataset, GoogleNet, AlexNet, LeNet.
- When evaluating the results I would recommend to use typical set of metrics (key performance indicators) for pattern recognition and machine learning.
- It would be worth to comment computational complexity of the algorithm 1.
- Conclusions and Future Research section is rather repetition of the abstract. Please add concluding remarks based on the result achieved.
Reviewer 3 Report
The authors proposed an interesting Spherically Stratified Point Projection approach for classifying points of interest in point clouds. But its comparison with other existing approaches has significant gaps.
There are a number of significant comments on the article:
1) The problem statement is unclear, apparently the authors are solving the problem of segmentation of LiDAR point clouds. It must be formulated explicitly, indicating what data is used at the input of the approach, what data is expected at the output. What category of methods is being investigated?
2) The article does not contain any mention of state-of-the-art methods for classifying point clouds: PAConv, GDANet, PointNet++, etc. It also does not contain modern methods of LiDAR point cloud segmentation: SPVNAS, Cylinder3D, KPConv, RPVNet, SalsaNext, etc.
3) In this regard, the review of existing works and the experimental part should be substantially expanded. Tables 3 and 4 should be significantly expanded.
4) The proposed method for projection of point clouds onto a sphere seems computationally complicated and is most likely calculated very slowly. The speed of the method is not noted in the article. The practical usefulness of the proposed approach seems questionable.
Round 2
Reviewer 1 Report
The paper was well revised. Still some suggestions:
- It's still suggested to combine Fig.1 and Fig.4 as fig.1 (a) and (b). And introduce them in one section so as to reduce some redundent explanation. Please consider to re-structure the paper.
- In Fig2, about the neighboring determination, I suggest to use some equation definition to make it more scientific.
- Please make more introduction about your KNU dataset. There are also some minority dataset for object classification, ex) Classification Using Hough Space of LiDAR Point Clouds. Please make some survey about these related researches.
Author Response
Thank you to the reviewers for giving us the opportunity to revision. Responses to comments are in the attachments.
Reviewer 2 Report
I am satisfied with the changes introduced to the revised version of the manuscript.
Author Response
thank you. The reviewer's comments helped a lot with our research.
Reviewer 3 Report
The authors have made significant improvements to their article in accordance with the comments.
At the same time, the authors should make a number of improvements in the presentation of the results obtained:
1) in Table 1 (Classification accuracy of previous studies using KNU dataset) and Table 4 (Evaluation of classification performance of previous studies) authors should add the results of the proposed sP^2 approach
2) in Table 3 (Classification performance evaluation of sP2), the authors should add an ablation study of the proposed approach in different modes / with different parameters.
After elimination of these remarks, the article can be published.
Author Response

(The authors gave the same response as above.)
